# Damage Evolution Modelling for Rock Materials Based on the Principle of Least Energy Dissipation Rate within Irreversible Thermodynamics

**DOI:** 10.3390/e24081032

**Published:** 2022-07-27

**Authors:** Xiaoming Li, Mingwu Wang

**Affiliations:** 1School of Civil and Hydraulic Engineering, Hefei University of Technology, Hefei 230009, China; hfutmingxiao@126.com; 2School of City and Architecture Engineering, Zaozhuang University, Zaozhuang 277160, China

**Keywords:** damage, irreversible thermodynamics, energy dissipation rate, constitutive relation, elastic–plastic

## Abstract

The nonlinear mechanical behavior of rock significantly influences the design and construction of underground structures. Due to the complexity and diversity of the damage mechanisms of rock, the damage variable directly defined by partial-damage mechanisms is insufficient in reflecting the progressive-failure process of rock comprehensively. So, in this paper, a novel damage variable is introduced into the plastic-strain rate based on the theoretical framework of irreversible thermodynamics to overcome this defect. The general expression is derived according to the least energy dissipation rate principle. The proposed damage variable can represent the irreversible energy dissipation process and has a strictly theoretical basis in mechanics. Moreover, the granite and marble stress-strain curves are simulated and compared with the Lemaitre damage model, Mazars damage model, and statistical damage model. The results show that the form of the proposed damage variable is practical and straightforward and can better reflect the entire stress-strain relationship of rock. Furthermore, the initial value of the inelastic response strain can be given directly through the proposed damage variable. The model presented here can overcome the issue that the current models need to select the damage threshold indirectly or assume it in advance and ensures that the damage evolution characteristics follow the first principle entirely.

## 1. Introduction

The nonlinear mechanical behavior of rock has been an essential issue of geomechanics for a long time. Many scholars have explored various constitutive models based on multiple mechanical and mathematical theories to reveal the nonlinear mechanical characteristics of stone [1,2,3,4,5,6,7,8]. However, the mechanical properties of the rock are affected by the formation environment, chemical element composition, stress history, and so on. No perfect model exists to capture the complex relationship between rock deformation and external load.

In recent decades, the continuous-damage mechanics have shone brilliantly in studying a constitutive model for rocks. Based on the continuous-damage theory, some damage models were built to describe the constitutive relation of rocks by mathematical expressions containing no physical meaning parameters [9,10,11,12,13]. Those models are called empirical models. On the other hand, Krajcinovic and Silva [14] were the first to combine the continuous-strength damage theory with statistics to study the mechanical behavior of rocks. Subsequently, some improved statistical damage models were constantly proposed with the statistical microscopic damage mechanics [15,16,17,18,19,20,21]. The key to characterizing the damage evolution in statistical damage theory is establishing a probability density function of the micro-elements’ strength. However, various damage variables are obtained if other strength criteria are adopted for the stress-strain curve, and even negative values will appear despite the same rock sample being used. In addition, the elastic modulus of the damaged and undamaged parts for statistical damage models is generally assumed to be consistent. Albeit the stresses are inconsistent, the strains remain the same. These do not conform to the physical equation in elastic mechanics. Then, based on the irreversible thermodynamics, there is another possibility of analyzing the damage mechanism of rock from the perspective of energy dissipation.

Unlike the statistical damage models, which rely on statistical physics to describe the failure probability property of micro-elements’ strength, there is a lack of first principles directly related to the failure probability of micro-elements’ strength. The constitutive equations obtained from irreversible thermodynamics satisfy the fundamental laws of thermodynamics. They describe the internal and external reactions of materials and characterize the irreversible energy dissipation process inside materials and the thermodynamic state of the materials system. They have a profound thermodynamic context and strict theoretical basis [22,23,24]. Some scholars have investigated the response law between rock deformation and external load by irreversible thermodynamics [25,26,27,28,29]. The above studies also indicate that the mechanical failure mechanisms of rock are multifaceted: existing micro-cracks slip, micro-pores fragmentation, elastic failure of mineral particles, dislocation movement of crystals, the development speed and direction of cracks, and the cluster effect between cracks or pores. These mechanisms are both the source of rock damage and the cause of energy dissipation. Thus, the concept of damage has been introduced to reveal this relationship better. Determining the quantitative relationship correctly between energy dissipation and the damage variable plays a vital role in constructing thermodynamic constitutive models.

Recently, applications of damage mechanics to irreversible thermodynamic models have been roughly classified into two categories. First, damage variables are not obtained from the scope of irreversible thermodynamics, but from other means. Wang et al. [30] used the Mazars damage criterion to define the damage variable. Darabi et al. [31] and Subramanian and Mulay [32] gave the damage variable based on the effective bearing area. Using statistical methods, Li et al. [33] directly defined the damage variable. Qu et al. [34] derived the Lemaitre damage variable by the fractional derivative. Although the mechanical evolution equations used are closely related to thermodynamics, the rationale of introducing damage variables directly into them seems to require more explanation and exploration. Second, damage variables have been studied and analyzed from different perspectives based on the concept of a non-equilibrium state in thermodynamics. Zhang [35] presented a damage-constitutive equation based on irreversible thermodynamics, but the equation did not explicitly express the damage variable. Based on the damage and flow criterion, Chai [36] deduced the damage variable in the framework of the elastic–plastic theory. He et al. [37] introduced the damage function potential and built empirical expressions of the damage variable related to temperature and material property. The latest research is that Gao et al. [38] deduced the damage evolution equation through the energy dissipation rate. Regrettably, the “dissipated energy” in that equation is the so-called effective elastic strain energy but is not the energy associated with unrecoverable strain in the process of rock deformation, and the constitutive evolution equation cannot describe the stress-strain relation after the stress peak.

In summary, although these studies were carried out within the framework of thermodynamics, some directly use the damage variable as an external introduction variable without deriving it from the energy perspective, or some depict the damage variable only from one or part of one damage mechanism. To overcome these shortcomings, this study combines the principle of least energy dissipation rate with elastic–plastic mechanics and further obtained the specific expression of the damage variable through rigorous mechanical derivation. The damage variable can comprehensively reflect the damage mechanism of rock and provide a reliable theoretical basis for subsequent study of the mechanical nonlinear behavior of the rock.

This paper introduces a new damage variable to the plastic strain rate. The specific expression of the proposed damage variable is derived based on the least energy dissipation rate principle. Furthermore, the equivalence relationship between the energy-dissipation rate and the damage energy-release rate is proved by introducing the damage variable into the free Helmholtz energy through the second principle of thermodynamics. In addition, the stress-strain relationship before the peak value of granite and the stress-strain relationship during the whole process of marble are simulated and verified, and the analytical and numerical solutions are further given. Finally, the proposed damage variable is further analyzed and discussed by comparing it with other theoretical models.

## 2. Damage Evolution

### 2.1. Theoretical Framework

The definition and derivation of the damage variable are the basis for analyzing the mechanical behaviors of solid materials by using continuous-damage mechanics, and as far as damage variables describing rock material damage are concerned, they can be divided into time-dependent and time-independent isotropic and anisotropic, etc. [5,6]. Selecting damage variables is essential in researching the damage evolution law of solid materials with irreversible thermodynamics.

The second principle of thermodynamics can be described as [39]:(1)$$\sigma :\dot{\varepsilon }-\varpi =\rho \dot{u}-T\dot{\eta }$$
where **σ** and $\dot{\varepsilon }$ are the second stress tensor and second strain rate tensor, respectively. The font is bold for tensors. *ϖ* is the spontaneous energy dissipation per unit mass. *u* is the internal energy per unit mass. *η* is the entropy per unit mass. ρ is the mass density. *T* is the absolute temperature. As one of the expressions of the second principle of thermodynamics, *ϖ* is required to be a negative value, 0 in reversible processes, and positive in irreversible processes. Herein, the free Helmholtz energy *ψ* can be defined as:(2)ψ=u−Tη

The increment of *ψ* can be written as:(3)$$\dot{\psi }=\dot{u}-T\dot{\eta }-\eta \dot{T}$$

By introducing Equation (3) into Equation (1), the expression for the second principle of thermodynamics in terms of the free Helmholtz energy can be written as:(4)$$\sigma :\dot{\varepsilon }-\varpi =\rho \dot{\psi }-\eta \dot{T}$$

If the temperature keeps constant, the above equation can be simplified as:(5)$$\sigma :\dot{\varepsilon }-\varpi =\rho \dot{\psi }$$

Helmholtz proposed the principle of least energy dissipation for incompressible viscous fluids based on irreversible thermodynamics. However, strict application conditions limit the application scope of this principle [40]. By introducing the concept of an instantaneous stable state, in the case of the linear and nonlinear non-equilibrium state, at any moment of any energy dissipation process, the system will take the minimum value of all possible energy dissipation rates under corresponding conditions. This modified energy dissipation principle is the principle of least energy dissipation rate [38,40,41]. The general form of the energy dissipation rate Φ is given as [41]:(6)$$\Phi =\int_{V}T\rho \frac{d_{i}\eta }{dt}dV$$
where *V* is the volume of material. *t* stands for time.

### 2.2. Damage Variable

Suppose a rock mesoscopic element is in a closed system in the deformation and failure process. In that case, there is no energy dissipation such as heat exchange and acoustic emission with the outside world. At this point, the work *U* done by external loading on the closed system is all converted into elastic energy *U**_e_* and dissipation energy *U**_d_*. The quantitative relationship is depicted as shown in Figure 1.

It is assumed that rock deformation is divided into elastic response deformation and inelastic response deformation under external loading, while inelastic response deformation is mainly composed of plastic deformation. In other words, plastic deformation is rock’s only energy dissipation mechanism in irreversible thermodynamics. As shown in Figure 1, the dissipation energy *U**_d_* is the work done by external loading on the plastic strain. Then, the energy dissipation rate *φ* of rock can be written as:(7)$$\varphi =\sigma :{\dot{\varepsilon }}^{p}$$
where, ${\dot{\varepsilon }}^{p}$ represents the plastic strain rate. The least energy consumption rate principle requires the least energy dissipation rate under the constraint condition of *g*(**σ**) = 0. The Lagrange multiplier *λ* is introduced, and there is the following expression:(8)$$\partial \left[\sigma :{\dot{\varepsilon }}^{p}-\lambda g(\sigma )\right]/\partial \sigma =0$$
where *g*(**σ**) is the constraint function, rearrange the above equation appropriately, and the following expression can be obtained:(9)$${\dot{\varepsilon }}^{p}=\lambda \partial g(\sigma )/\partial \sigma$$

Equation (9) can be further written as an increment of the plastic strain **ε****^p^** as:(10)$$d\varepsilon^{p}=d\lambda \partial g(\sigma )/\partial \sigma$$

As we all know, the plastic potential theory has been widely used in plastic mechanics, and Equation (10) happens to be the result. Like Drucker’s postulate, the principle of least energy dissipation rate can also be used as one of the theoretical bases of plastic potential theory. At this point, *g*(**σ**) can be called the plastic potential function. The properties of materials affecting the energy dissipation process will be determined in a specific form, namely the corresponding constraint conditions. When the above constraints are the yield conditions, the result of the plastic potential theory is a flow law associated with the loading conditions. Otherwise, it is called uncorrelated flow.

According to the incremental theory of plastic mechanics [42,43], the plastic strain rate is expressed by
(11)$${\dot{\varepsilon }}_{i}^{p}=\frac{2\alpha }{3}\left[\sigma_{i}-\frac{1}{2}(\sigma_{j}+\sigma_{k})\right]$$
where *i*, *j*, and *k* rotate in the order of 1, 2, and 3; *α* is a non-negative scalar constant of proportionality. In general, two state variables, strain and damage can describe materials’ mechanical damage behaviors. The formation and development of micro-pores and micro-cracks in the internal structure of the materials cause irreversible damage to the materials. It can be considered that this irreversible damage is the leading cause of inelastic response strain.

Based on the background of isotropy, the plastic strain is assumed here to be positively correlated with the damage variable *D*. As can be seen from Equation (11), when the three principal stresses remain unchanged, the plastic strain rate maintains a linear growth with *α*. However, in most cases, the plastic strain rate increases nonlinearly, coinciding with the nature of the damage variable, so the damage change rate is used to replace the constant α. In addition, take the principal axis of stress as the coordinate axis, combined with the effective stress principle in damage mechanics, by blending Equation (7) with Equation (11), the energy dissipation rate φ can be obtained as:(12)$$\varphi =\sigma :{\dot{\varepsilon }}^{p}=\frac{2\dot{D}(t)}{3\left[1-D(t)\right]}(\sigma_{1}^{2}+\sigma_{2}^{2}+\sigma_{3}^{2}-\sigma_{1}\sigma_{2}-\sigma_{2}\sigma_{3}-\sigma_{1}\sigma_{3})$$
where *D*(*t*) represents the damage variable at any time, and $\dot{D}(t)$ is the derivative concerning time *t*.

Take the Mises criterion as the constraint:(13)$$\sigma_{1}^{2}+\sigma_{2}^{2}+\sigma_{3}^{2}-\sigma_{1}\sigma_{2}-\sigma_{2}\sigma_{3}-\sigma_{1}\sigma_{3}=\sigma_{s}^{2}$$
where *σ**_s_* is the yield limit.

The plastic strain rate containing damage can be obtained by substituting Equations (12) and (13) into Equation (8) as:(14)$$\frac{2\dot{D}(t)}{3\left[1-D(t)\right]}(2\sigma_{i}+\sigma_{j}+\sigma_{k})+\lambda (2\sigma_{i}+\sigma_{j}+\sigma_{k})=0$$

Solve Equation (14) by integrating, the expression of *D*(*t*) can be obtained as:(15)$$D(t)=1-e^{-\frac{3}{2}\lambda t+c}$$

It can be seen from Equation (14) that the damage variable obtained here applies to both uniaxial and triaxial tests of materials. In the loading experiment of constant strain rate, assuming that the strain is linearly correlated with the constant *k*, then
(16)ε=kt

To further analyze the stress-strain relationship of rock, Equation (16) is substituted into Equation (15), and the expression of *D* can be written as
(17)$$D=1-\exp(-\frac{3}{2}\beta \varepsilon +c)$$
with
(18)$$\beta =\frac{\lambda }{k}$$
where *β* and *c* can be regarded as the dimensionless parameters introduced in the mathematical derivation obtained through experiments. The damage variable *D* here applies to both rock compression and tensile tests.

### 2.3. Damage Threshold and Parameter Determination

The continuous damage-constitutive equation for rocks [39] can be expressed as
(19)$$\sigma =(1-D)\left[\lambda_{0}tr\varepsilon +2G_{0}\varepsilon \right]$$
where *λ*_0_ and *G*_0_ are the Lame coefficients.

The initial stage of compression of rocks is usually called the elastic compaction stage. In other words, at this position, the micro-holes or micro-cracks of the internal rock structure are compressed, and no damage occurs at this stage [44]. In the stress-strain curve of rock, when the stress gradually approaches the peak value, the elastic energy of rock increases, and the irreversible dissipated energy increases. The plastic strain begins to appear, and the damage is thought to have started [45]. Take the total strain value when the plastic strain occurs as the threshold of elastic–plastic damage, and the constitutive damage equation can be written as flows
(20)$$\sigma =\left\{ \begin{array}{cc} \lambda_{0}tr\varepsilon +2G_{0}\varepsilon & for\;\varepsilon \leq \;\varepsilon_{0}\\ (1-D)\left[\lambda_{0}tr\varepsilon +2G_{0}\varepsilon \right] & for\;\varepsilon >\varepsilon_{0}\; \end{array}\right.$$
where **ε**_0_ is the total strain value when the plastic strain occurs. In the uniaxial compression experiment, the stress is
(21)$$\left\{ \begin{array}{l} \sigma =\sigma_{1}\\ \sigma_{2}=\sigma_{3}=0 \end{array}\right.$$

Combined with Equations (17), (20) and (21), the constitutive model of rock under uniaxial compression is described as
(22)$$\sigma =E\varepsilon \exp(-\frac{3}{2}\beta \varepsilon +c)$$

Then, material parameters can be determined by the specific rock stress-strain curve. For example, in the axial direction:(23)$$\left\{ \begin{array}{cc} \sigma =\sigma_{\max} & for\;\varepsilon =\varepsilon_{m}\\ d\sigma /d\varepsilon =0 & for\;\varepsilon =\varepsilon_{m} \end{array}\right.$$
where, *ε**_m_* and *σ*_max_ are the strain and stress at the peak of the stress-strain curve, respectively.

Substituting Equation (22) into Equation (23), there is the following equation
(24)$$E\varepsilon_{m}\exp(-\frac{3}{2}\beta \varepsilon_{m}+c)(1-\frac{3}{2}\beta \varepsilon_{m})=0$$To solve the parameter *β*, let
(25)$$(1-\frac{3}{2}\beta \varepsilon_{m})=0$$Here, the value of *β* can be obtained as
(26)$$\beta =\frac{2}{3\varepsilon_{m}}$$

By Equations (22), (23) and (26), the value of parameter *c* can be calculated as follows
(27)$$c=1+\ln\sigma_{\max}-\ln E\varepsilon_{m}$$

The above analysis shows that rock damage occurs only in response to plastic strain, not at the elastic compaction stage. If the damage variable is 0, according to Equation (17), the following formula can be obtained as
(28)$$D=1-\exp(-\frac{3}{2}\beta \varepsilon +c)=0$$

Solving Equation (28), the initial value of inelastic response strain *ε*_0_ (the strain damage threshold) under uniaxial compression can be obtained as
(29)$$\varepsilon_{0}=\frac{2c}{3\beta }\;for\;\varepsilon_{0}<\varepsilon_{m}$$

In addition to the above parameter determination method, the nonlinear fitting process can also be adopted.

### 2.4. Damage Energy Release Rate

The damage energy release rate [26,27,34] can be expressed as
(30)$$\phi =-\rho \frac{\partial \psi }{\partial D}$$
where *ϕ* is the damage energy release rate. *Ψ* still represents the free Helmholtz energy, but as a function of the damage variable *D* and the strain tensor **ε**, which can usually be written as
(31)ψ=ψ(ε,D)

From Equation (31), *ψ* only has one term. In some studies, the function *ψ* consists of two terms [23,26]. It is considered that this study is limited to the elastic–plastic theoretical framework. The function of the free Helmholtz energy can be assumed as
(32)$$\psi =\psi_{0}(\varepsilon )-\psi_{1}(\varepsilon ,D)$$

Equation (32) is an implicit expression. The precise form of the process is unknown. The decision of the particular function expression of *ψ* is problematic, primarily depending on the need for research. Herein, the damage energy release rate *ϕ* can be obtained from Equations (30) and (32), namely,
(33)$$\varphi =\rho {\psi^{\prime }}_{1}(\varepsilon ,D)=\rho \frac{\partial \psi_{1}(\varepsilon ,D)}{\partial D}$$

Combined with the above analysis, by comparison with Equation (5), since plastic strain energy is the only energy dissipation mechanism, ϖ is also a function of the strain tensor **ε** and damage variable *D*, and Equation (32) can be further written as
(34)$$\varphi =\rho \frac{\partial \psi_{1}(\varepsilon ,D)}{\partial D}=\dot{\varpi }(\varepsilon ,D)$$
where $\dot{\varpi }$ represents taking the derivative for damage variable *D*.

Assuming that the irreversible dissipation energy is the work done on the plastic strain, there is the following equation
(35)$$\varpi (\varepsilon ,D)=\sigma :\varepsilon^{p}$$

The derivative of Equation (35) concerning the damage variable is:(36)$$\dot{\varpi }(\varepsilon ,D)=\sigma :{\dot{\varepsilon }}^{p}$$

Rearrange Equations (7), (12), and (34)–(36) appropriately, and there is the following expression
(37)φ=ϕ

It can be seen from Equation (37) that the energy dissipation rates obtained are consistent whether the damage variable is directly introduced into the free Helmholtz energy or the mechanism of energy input and dissipation.

## 3. Validation of Damage Constitutive Model

### 3.1. Application in Granite

The rationality of the damage variable proposed here needs to be verified by experimental data. Gao et al. [38] performed uniaxial compression and uniaxial tensile experiments on granite from the Three Gorges Project in China. They applied the experimental data obtained to the relevant rock-constitutive model. The date was also used in this study. For brittle rocks, the Mazars damage model (MDM) can describe its stress-strain relationship well and is outstanding among many constitutive damage models, and its damage variable can be expressed as
(38)$$D=\left\{ \begin{array}{cc} 0 & for\;\varepsilon^{*}\leq \varepsilon_{p}^{*}\\ 1-\frac{\varepsilon_{p}^{*}(1-\lambda_{1})}{\varepsilon^{*}}-\frac{\lambda_{1}}{\exp\left[\lambda_{2}(\varepsilon^{*}-\varepsilon_{p}^{*})\right]} & for\;\varepsilon^{*}<\varepsilon_{p}^{*} \end{array}\right.$$

with (39)$$\varepsilon^{*}=\sqrt{{\left(\langle \varepsilon_{1}\rangle +\right)}^{2}+{\left(\langle \varepsilon_{2}\rangle +\right)}^{2}+{\left(\langle \varepsilon_{3}\rangle +\right)}^{2}}$$
where $\varepsilon_{p}^{*}$ is the maximum value of *ε*. *Λ*_1_ and *λ*_2_ are parameters that must be obtained by fitting according to the actual stress-strain curve.

Therefore, the proposed model is compared with the Mazars damage model and the damage model in Gao et al. [38] to verify the rationality of the new model. Mechanical parameters of granite are listed in Table 1.

Equations (26) and (27) determine the values of the dimensionless parameters β and c. In addition, the fitting results of parameters *λ*_1_ and *λ*_2_ for the Mazars damage model in Gao et al. [38] are adopted. The calculation results of all parameters are shown in Table 2.

The comparison of simulated values obtained from three models with experimental data is shown in Figure 2. The mean relative errors of the calculated values of the three models are shown in Table 3 (where EDDM is the energy dissipation damage model in Gao et al. [38], and NEDDM is the new energy dissipation damage model with the damage variable proposed here).

As shown in Figure 2, three models can describe granite’s stress-strain relationship. As a hard, brittle rock, the elastic modulus of granite is more significant than other rocks. However, the curve trend, the stress-strain relationship of granite, was close to a straight line in the elastic stage. After entering the plastic phase (*ε* > *ε*_0_), the damage will increase rapidly, resulting in a brittle failure after the rock strength rises to the peak value. After that, the rock’s strength decreases rapidly from the peak value, and finally, the rock will lose stability.

In the uniaxial compression case, as shown in Table 2, the strain damage thresholds were *ε**_0_* = 5.98 × 10^−4^ < *ε*_m_ = 7.33 × 10^−4^, and this is consistent with the view of classical continuous damage mechanics that rock damage has occurred before the peak value [44]. The same situation exists in the constitutive model of uniaxial tension. In the elastic response stage, the calculated curves of the three models were nearly straight lines and close to the experimental curve. In section OA, the stress level increased gradually, and the rock’s internal structure produced micro-cracks, and damage appeared and developed accordingly. The curve trends of the three constitutive models were still consistent, but the Mazars damage model calculation curve increased by a more considerable margin. However, in section AB, although the trends of the simulated curves fluctuated to some extent, they all turned downward at the end. In terms of the fitting degree with the experimental curve, the calculated curve obtained from the proposed model was closer, which can be found in Table 3. The mean relative errors of the proposed model were less than the other two. The performance of the three constitutive models in the uniaxial tensile experiment was the same as that in the uniaxial compression experiment, so it will not be repeated here.

### 3.2. Application in Marble

The above example preliminarily tested the ability of the new model to describe the stage before the stress peak of granite under uniaxial compression. To further verify the new model’s applicability, marble’s whole stress-strain relationship was simulated and analyzed. Rummel and Fairhurst [22] performed uniaxial and conventional triaxial compression experiments on Tennessee marble samples. Based on these experimental data, Li et al. [16] and Li et al. [17] verified the rationality of their respective statistical damage-constitutive model. These experimental data continued to be used to verify the proposed model. A comparative analysis was also conducted with these two statistical damage-constitutive models.

The core theoretical basis of the statistical damage-constitutive model of rock is the assumption that the rock is composed of a series of micro-elements. The strengths of all micro-elements obey random distribution. Li et al. [16] assumed that the micro-elements’ strengths of rock follow the Weibull distribution function. In contrast, Li et al. [17] believed that the micro-elements’ strengths of rock obey the maximum entropy distribution function, and the specific damage variable expressions are as follows:(40)$$D_{w}=1-\exp\left[-{\left(\frac{F}{F_{0}}\right)}^{m}\right]$$
(41)$$D_{s}=\int_{0}^{n}\left[\exp(\lambda_{0}\sum_{j=1}^{n}\lambda_{j}u_{j}(x)\right]dx$$
where *D**_w_* and *D**_s_* represent the damage variables defined by the Weibull and the maximum entropy, respectively. *F* is the strength of the micro-element. *m* is the coefficient related to the shape. *F*_0_ is the scale parameter. *x* is the statistical variable, which mainly refers to the strength of the micro-element. *j* is the number of Lagrange multipliers. *u_j_*(*x*) is the function related to the moment. *λ* is the Lagrange multiplier. *D**_w_* and *D**_s_* tend to go from 0 to 1. The mechanical parameters of marble from Rummel and Fairhurst [22] are shown in Table 4.

Due to the complex stress-strain relationship in the whole deformation process of marble, the description ability of Equation (20) is insufficient, and the damage-constitutive equation proposed by Zhao et al. [18] is adopted as
(42)$$\sigma_{1}=E\varepsilon_{1}(1-D)+2\nu \sigma_{3}+(1-2\nu )R_{c}D$$
(43)$$R_{c}=\frac{2c_{r}\cos\varphi_{r}+\sigma_{3}(1+\sin\varphi_{r})}{1-\sin\varphi_{r}}$$
where *R**_c_* is the residual strength. *c**_r_* and *φ**_r_* represent the residual cohesion and residual internal friction angle. *D* is calculated by Equation (17). Let *R* be the complex correlation coefficient, and the final calculation results of parameters in the proposed model are shown in Table 5.

The comparison of three calculated curves with experimental curves for marble is demonstrated in Figure 3. The mean relative errors of the calculated values of the three models are shown in Table 6. Figure 4 shows the comparison between the damage variables in the proposed model and Li et al. [17] under different confining pressures (where SDM is the statistical damage model in Li et al. [16], MESDM is the maximum entropy statistics damage model in Li et al. [17]. NEDDM is the new energy dissipation damage model with the damage variable proposed here).

On the one hand, as shown in Figure 3, the three theoretical models fully reflected the mechanical characteristics of elastic compaction, strain hardening, and strain softening for marble, especially for the curve before the stress peak. When loading was over the stress peak, the phenomenon of strain-softening began to appear with the gradual increase in damage. At this point, the simulation accuracies of the three theoretical models started to decline to vary degrees. All deviated from the experimental curve more or less but developed along the track of strain-softening on the whole. As shown in Table 6, under different confining pressures, the mean relative errors of the proposed model were less than ten percent, and the total mean relative error was the smallest.

On the other hand, the proposed model gave the critical values of elastic–plastic strain for marble under different confining pressures, shown in Table 5 and Figure 4. It is easy to see that the critical values *ε*_0_ were less than the strain *ε*_m_ at the stress peak [9,10,11,46]. This is also consistent with the viewpoint of classical continuous damage mechanics.

Although the damage threshold was also set in the other two models, critical values of the elastic and plastic strains could not be given. As shown in Figure 4, the damage variables proposed here and in Li et al. [17] changed on a scale of zero to one, but the corresponding initial strain values differed. The damage variables in Li et al. [17] began to appear to occur at the moment of loading, but, from Lemaitre and Desmorat [47], usually, during the elastic phase, only the existing internal micro-cracks and micro-pores themselves were compressed, and no actual damage occurred. Another difference is that the change rate of the proposed damage variable gradually decreased, and the damage value approached one eventually. Still, the trend of the change rate for the damage variable in Li et al. [17] showed a state of non-convergence.

The other thing to note is that the three theoretical models differed from the experimental values. The possible reason is that the mechanical behavior of rock is highly complex, while the three damage theoretical models are essentially phenomenological mathematical models. After all, describing such a complex stress-strain relationship is a challenging topic.

## 4. Conclusions

A rational damage model is critical for characterizing the nonlinear deformation of rock. Although statistical damage models can describe nonlinear mechanical behavior, the proposed damage variable in these models must assume the distribution function form of micro-elements’ strength. A novel damage variable was derived here by the least energy dissipation rate principle based on irreversible thermodynamics to remedy this drawback. Relative to statistical damage variables, the damage evolution method presented here was tested using granite and marble experimental data, and comparisons were further conducted with statistical damage evolution theory. The conclusions of the current work are as follows.

(1)In irreversible thermodynamics, the damage variable with a simple form is derived by the least energy dissipation rate principle. By introducing the proposed damage variable into the free Helmholtz energy as an internal variable, the equivalence relationship between the energy dissipation rate and the damage energy release rate was proved by the second principle of thermodynamics. The rigorous thermodynamic theory was the cornerstone of this research. The proposed damage evolution method can represent the energy dissipation of rock during the deformation and has good universality.(2)Through the simulations of the stress-strain relationship of granite and marble, the calculated values obtained by the proposed model were close to the experimental values, and the fitting accuracy was slightly higher than other current models. The proposed model is applicable if the stress-strain relationship is simulated before the stress peak or the whole process. Parameter calculation is more convenient and concise.(3)The relationship between the new damage variable and strain was calculated, summarized, and compared with the current statistical damage variable. The results show that the evolution trends of the two were the same, but the former could give the critical value between elastic and plastic strain, while the latter could not. Energy dissipation is the first principle directly related to damage. The strain damage threshold obtained through rigorous deduction is more scientific and reasonable than the artificial assumed damage threshold.(4)This paper mainly focused on the theoretical damage evolution method and only applied to isotropic damage. Furthermore, verification of other rocks’ mechanical damage evolution characteristics is required. Future research should integrate the suggested damage evolution into the numerical simulation.

## Figures and Tables

**Figure 1 entropy-24-01032-f001:**
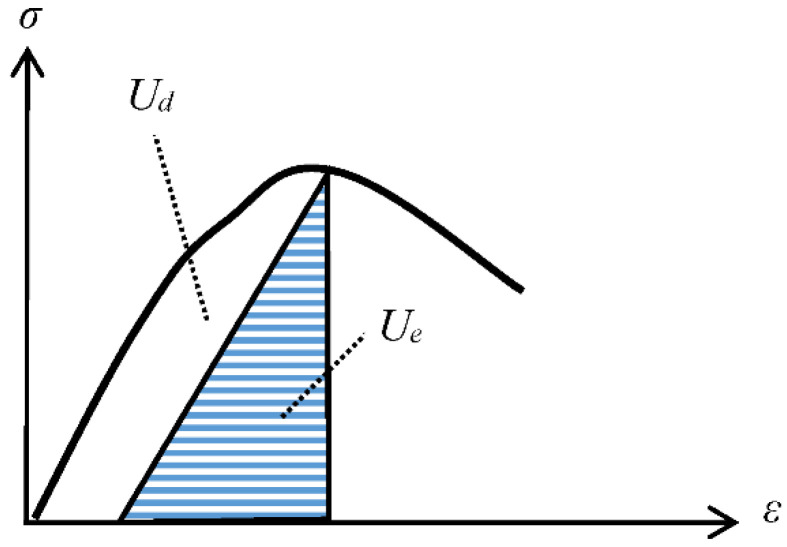
Quantitative relationship of dissipation energy and elastic energy.

**Figure 2 entropy-24-01032-f002:**
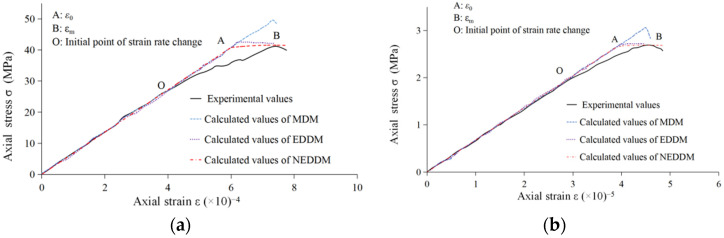
Comparison of three theoretical models with experimental curves: (**a**) uniaxial compression; and (**b**) uniaxial tensile.

**Figure 3 entropy-24-01032-f003:**
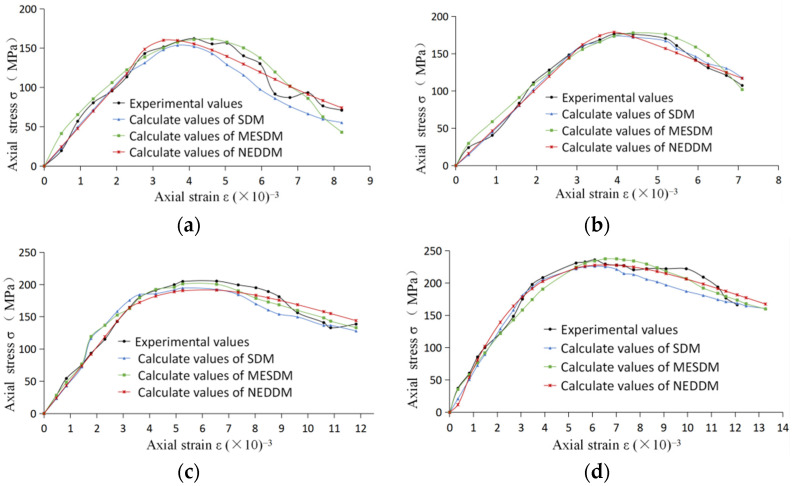
Comparison of three calculated curves with experimental curves at different confining pressures: (**a**) *σ*_2_ = *σ*_3_ = 3.5 MPa; (**b**) *σ*_2_ = *σ*_3_ = 7 MPa; (**c**) *σ*_2_ = *σ*_3_ = 14 MPa; and (**d**) *σ*_2_ = *σ*_3_ = 21 MPa.

**Figure 4 entropy-24-01032-f004:**
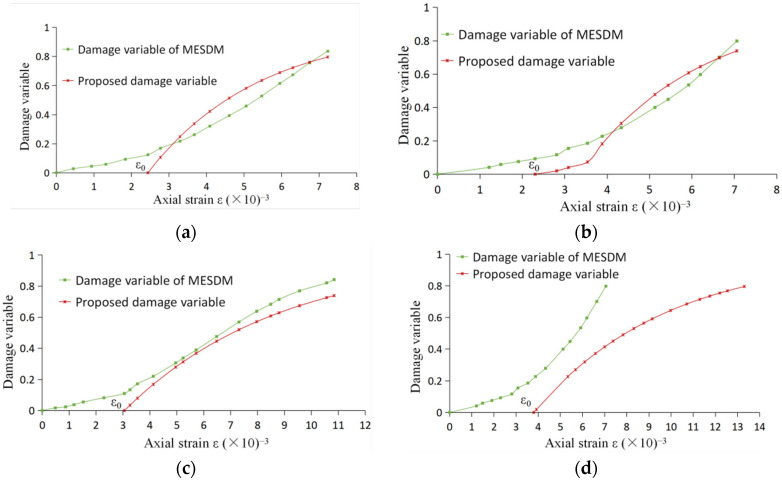
Comparison of the damage variables in this paper and Li et al. [17]: (**a**) *σ*_2_ = *σ*_3_ = 3.5 MPa; (**b**) *σ*_2_ = *σ*_3_ = 7 MPa; (**c**) *σ*_2_ = *σ*_3_ = 14 MPa; and (**d**) *σ*_2_ = *σ*_3_ = 21 MPa.

**Table 1 entropy-24-01032-t001:** Mechanical parameters of granite specimen from Gao et al. [38].

Elastic Module (*E*)	Poisson’s Ratio (*ν*)	Cohesion(*c*)	Internal Friction Angle (*φ*)	Uniaxial Compressive Strength (*σ*_max_)	Uniaxial Tensile Strength (*σ*_max_)
68 GPa	0.24	31 MPa	48.47°	41.5 MPa	2.691 MPa

**Table 2 entropy-24-01032-t002:** Parameters of the proposed model and the comparison model.

Experiments	*ε* _m_	*β*	*c*	*ε* _o_	*λ* _1_	*λ* _2_
Uniaxial compression	7.33 × 10^−4^	909.5	0.8168	5.98 × 10^−4^	1	23,290
Uniaxial tensile	4.5 × 10^−5^	14814.8	0.8714	3.92 × 10^−5^	1	124,110

**Table 3 entropy-24-01032-t003:** Mean relative errors of the calculated values of the three models for granite (%).

Models	Test		Total Mean Relative Error
	Uniaxial Compression	Uniaxial Tensile	
MDM	8.1	13.7	21.8
EDDM	12.5	10.6	23.1
NEDDM	5.1	5.6	10.7

**Table 4 entropy-24-01032-t004:** Mechanical parameters of marble [22].

Elastic Module (*E*)	Poisson’s Ratio (*ν*)	Cohesion (*c*)	Internal Friction Angle (*φ*)	Uniaxial Compressive Strength (*σ*_max_)
51.62 GPa	0.25	27.96 MPa	44.01°	123 MPa

**Table 5 entropy-24-01032-t005:** Parameters of the proposed model for marble.

σ_3_ (MPa)	ε_m_	β	δ	ε_0_	R_c_	R
3.5	4.13 × 10^−3^	221.6	0.81	2.4 × 10^−3^	69.8	0.98
7	4.3 × 10^−3^	240.3	1.19	2.6 × 10^−3^	102.3	0.96
14	6.5 × 10^−3^	115.0	0.52	3.0 × 10^−3^	125.6	0.96
21	7.0 × 10^−3^	11.78	0.63	3.8 × 10^−3^	155.4	0.98

**Table 6 entropy-24-01032-t006:** Mean relative errors of the calculated values of the three models for marble (%).

Models	Confining Pressures/MPa				Total Mean Relative Error
	3.5	7	14	21	
MDM	12.6	6.4	9.7	7.8	36.5
EDDM	15.9	8.4	6	6.1	36.4
NEDDM	8.7	6.1	6.5	5.8	27.1

## Data Availability

Not applicable.

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
