# Peer review of "Damage Evolution Modelling for Rock Materials Based on the Principle of Least Energy Dissipation Rate within Irreversible Thermodynamics"

_entropy, 2022, doi:10.3390/e24081032_

Round 1

Reviewer 1 Report

The authors has introduced a new damage variable into the plastic strain rate, then they derived the specific expression of the proposed damage variable based on the principle of least energy dissipation rate to represent the irreversible energy dissipation process. The subject is relevant to the scope of the journal and the paper is very well organized. The work is original and there is a significant amount of new work in the paper. This study is worthy of publication. The paper is recommended for publication after addressing the following comments:

-        - Please calculate and present percentage differences for the comparisons carried out in this manuscript.

Author Response

< Manuscript Number: entropy-1799086 >

< Entropy >

< Damage evolution modelling for rock materials based on the principle of least energy dissipation rate within irreversible thermodynamics >

Dear Reviewer,

Thank you for your helpful comments and suggestions on our manuscript.

We have modified the manuscript accordingly, and the modified part is marked in colored red fonts in the current version of the manuscript.

Detailed corrections are listed below point by point

Reviewers' comments:

Reply to Reviewer 1

Reviewer #1: The authors has introduced a new damage variable into the plastic strain rate, then they derived the specific expression of the proposed damage variable based on the principle of least energy dissipation rate to represent the irreversible energy dissipation process. The subject is relevant to the scope of the journal and the paper is very well organized. The work is original and there is a significant amount of new work in the paper. This study is worthy of publication. The paper is recommended for publication after addressing the following comments:

  1. Please calculate and present percentage differences for the comparisons carried out in this manuscript.

Thank you very much for your suggestions. We have calculated the mean relative errors of the computed values of different models, and the related analysis is added. The specific contents are as follows:

Page#8, Lines#319-323; Page#9, Lines#348-349; Page#10, Lines#395-402; Page#11, Lines#417-420 (Revised version):

The comparison of simulated values obtained from three models with experimental data is shown in Fig 2. The mean relative errors of the calculated values of the three models are listed in Table 3. (Where MDM is the Mazars damage model. EDDM is the energy dissipation damage model in Gao et al. [38]. NEDDM is the new energy dissipation damage model with the damage variable proposed here)

Table 3. Mean relative errors of the calculated values of the three models for granite (%).

Models

Test

Total mean relative error

Uniaxial compression

Uniaxial tensile

MDM

8.1

13.7

21.8

EDDM

12.5

10.6

23.1

NEDDM

5.1

5.6

10.7

In the uniaxial compression case, as shown in Table 2, the strain damage thresholds ε0 =5.98×10-4<εm=7.33×10-4, this is consistent with the view of classical continuous damage mechanics that rock damage has occurred before the peak value [44], and the same situation exists in the constitutive model of uniaxial tension. In the elastic response stage, the calculated curves of the three models are nearly straight lines and close to the experimental curve. In section OA, the stress level increases gradually, and the rock's internal structure has produced micro-cracks, and the damage appears and develops accordingly. The curve trends of the three constitutive models are still consistent, but the calculation curve of the Mazars damage model increases by a more considerable margin. However, in section AB, although the trends of the simulated curves fluctuate to some extent, they all turned downward at last. In terms of fitting degree with the experimental curve, the calculated curve obtained from the proposed model is closer, which can be found in Table 3. The mean relative errors of the proposed model are less than the other two. The performance of the three constitutive models in the uniaxial tensile experiment is the same as that in the uniaxial compression experiment, so it will not be repeated here.

The comparison of three calculated curves with experimental curves for marble is demonstrated in Fig 3. The mean relative errors of the calculated values of the three models are shown in Table 6. Fig 4 shows the comparison between the damage variables in the proposed model and Li et al. [17] under different confining pressures. (Where SDM is the statistical damage model in Li et al. [16], MESDM is the maximum entropy statistics damage model in Li et al. [17], and NEDDM is the new energy dissipation damage model with the damage variable proposed here).

Table 6. Mean relative errors of the calculated values of the three models for marble (%).

Models

Confining pressures /MPa

Total mean relative error

3.5

7

14

21

MDM

12.6

6.4

9.7

7.8

36.5

EDDM

15.9

8.4

6

6.1

36.4

NEDDM

8.7

6.1

6.5

5.8

27.1

On the one hand, as shown in Fig 3, the three theoretical models can fully reflect the mechanical characteristics of elastic compaction, strain hardening, and strain softening for marble. Especially for the curve before the stress peak. When loading is over the stress peak, the phenomenon of the strain-softening begins to appear with the gradual increase of damage. At this point,the simulation accuracies of the three theoretical models began to decline to varying degrees,and all deviated from the experimental curve more or less, but developed along the track of strain-softening on the whole. As shown in Table 6, under different confining pressures, the mean relative errors of the proposed model are less than ten percent, and the total mean relative error is the smallest.

Reviewer 2 Report

In the present work a novel damage variable is introduced into the plastic strain rate based on the theoretical framework of irreversible thermodynamics. The proposed damage variable can represent the irreversible energy  dissipation process and has a strictly theoretical basis in mechanics. The results show that the form of the proposed damage variable is practical and straightforward and can better reflect the entire stress-strain relationship of rock. The work is well written and attract the interest of the readers. It could be accepted for publication. 

Author Response

< Manuscript Number: entropy-1799086 >

< Entropy >

< Damage evolution modelling for rock materials based on the principle of least energy dissipation rate within irreversible thermodynamics >

Dear Reviewer

Thank you for your helpful comments and suggestions on our manuscript.

Reviewers' comments:

Reply to Reviewer 2

Reviewer #2: In the present work a novel damage variable is introduced into the plastic strain rate based on the theoretical framework of irreversible thermodynamics. The proposed damage variable can represent the irreversible energy  dissipation process and has a strictly theoretical basis in mechanics. The results show that the form of the proposed damage variable is practical and straightforward and can better reflect the entire stress-strain relationship of rock. The work is well written and attract the interest of the readers. It could be accepted for publication.

Thank you for your review and affirmation.

Reviewer 3 Report

In this paper, a pretty old problem in the filed of mechanical engineering is investigated using a reliable methodology. However, the topic is very practical and industry oriented, and thus it would be interesting to potential readers. The results look solid and accurate as they have been compared with the experimental data in Figure 2, and a good match was found. Nevertheless, before publication of the paper, the following important issue needs to be addressed carefully: 1) What are/is the main novel aspect/aspects of the paper with respect to available and similar papers in the literature? Please add these aspects(this aspect) to the introduction section of the paper.

Author Response

< Manuscript Number: entropy-1799086 >

< Entropy >

< Damage evolution modelling for rock materials based on the principle of least energy dissipation rate within irreversible thermodynamics >

Dear Reviewer

Thank you for your helpful comments and suggestions on our manuscript.

We have modified the manuscript accordingly, and the modified part is marked in colored red fonts in the current version of the manuscript.

Detailed corrections are listed below point by point

Reviewers' comments:

Reply to Reviewer 3

Reviewer #3: In this paper, a pretty old problem in the filed of mechanical engineering is investigated using a reliable methodology. However, the topic is very practical and industry oriented, and thus it would be interesting to potential readers. The results look solid and accurate as they have been compared with the experimental data in Figure 2, and a good match was found. Nevertheless, before publication of the paper, the following important issue needs to be addressed carefully:

1) What are/is the main novel aspect/aspects of the paper with respect to available and similar papers in the literature? Please add these aspects(this aspect) to the introduction section of the paper.

Thank you very much for your suggestions. We have integrated and rewritten part of the "Introduction" to emphasize and clarify the main novel aspects of this paper.

Pages#2-3, Lines#71-101 (Revised version):

Recently, applications of damage mechanics to irreversible thermodynamic models are roughly classified into two categories. First, damage variables are not obtained from the scope of irreversible thermodynamics, but from other means. Wang et al. [30] used the Mazars damage criterion to define the damage variable. Darabi et al. [31] and Subramanian and Mulay [32] given the damage variable based on the effective bearing area. Using statistical methods, Li et al. [33] directly defined the damage variable. Qu et al. [34] derived the Lemaitre damage variable by the fractional derivative. Although the mechanical evolution equations used are closely related to thermodynamics, the rationale of introducing damage variables directly into which seems to require more explanation and exploration. Second, damage variables are studied and analyzed from different perspectives based on the concept of a non-equilibrium state in thermodynamics. Zhang [35] presented a damage constitutive equation based on the irreversible thermodynamics, but the equation did not explicitly express the damage variable. Based on the damage and flow criterion, Chai [36] deduced the damage variable in the framework of the elastic-plastic theory. He et al. [37] introduced the damage function potential and built empirical expressions of the damage variable related to temperature and material property. The latest research is that Gao et al. [38] deduced the damage evolution equation through the energy dissipation rate. Regrettably, the "dissipated energy" in that is the so-called effective elastic strain energy, but is not the energy which is associated with unrecoverable strain in the process of rock deformation, and the constitutive evolution equation cannot describe the stress-strain relation after the stress peak.

In summary, although these studies are carried out within the framework of thermodynamics, some directly use the damage variable as an external introduction variable without deriving it from the energy perspective, or some depict the damage variable only from one or part damage mechanisms. To overcome these shortcomings, this study combines the principle of least energy dissipation rate with elastic-plastic mechanics and further obtained the specific expression of the damage variable through rigorous mechanical derivation. The damage variable can comprehensively reflect the damage mechanism of rock and provide a reliable theoretical basis for the subsequent study of the mechanical nonlinear behavior of the rock.

Round 2

Reviewer 3 Report

The required revisions are made. Now the paper is recommended for publication.